# Spatial Structure Characteristics of Slope Farmland Quality in Plateau Mountain Area: A Case Study of Yunnan Province, China

**Zhengfa Chen [1,2] and Dongmei Shi [2,*]**

[1]    Kunming Engineering Corporation Limited of POWERCHINA, Kunming 650051, China; chenzhengfa2013@126.com

[2]    College of Resources and Environment, Southwest University, Chongqing 400715, China

*    Correspondence: shidm_1970@126.com; Tel.: +86-1388-3159-167

**Abstract:** As an important part of farmland, the slope farmland is widely distributed in the central and western plateau mountain region in China. It is necessary to scientifically evaluate the slope farmland quality (SFQ) and analyze the spatial structure characteristics of SFQ to ensure reasonable utilization and partition protection of slope farmland resources. This paper takes the typical plateau mountain region—Yunnan Province in China—as an example and systematically identifies the leading factors of SFQ. The sloping integrated fertility index (SIFI) is adopted to reflect the SFQ. The evaluation system is built to quantitatively evaluate the SFQ and the spatial structure characteristics of SFQ were analyzed by a geostatistical model, autocorrelation analysis and spatial cold–hot spot analysis. The results show that the SFQ indexes in Yunnan Province are between 0.36 and 0.81, with a mean of 0.59. The SFQ grade is based on sixth-class, fifth-class, seventh-class and fourth-class land. The SFQ indexes present a normal spatial distribution, and the Gaussian model fits well with the semi-variance function of the spatial distribution of SFQ indexes. Furthermore, the spatial distribution of SFQ indexes is moderately autocorrelated. The structural factors play a major role in the spatial heterogeneity of SFQ indexes, but the influence of random factors should not be ignored. The spatial distribution of SFQ grades has a significant spatial aggregation characteristic, and the types of local indicators of spatial association (LISA) are based on high–high (HH) aggregation and low–low (LL) aggregation. The cold spot and hot spot distributions of SFQ grades display the significant spatial difference. The hot spot area is mainly distributed in Central Yunnan and the Southern Fringe, while the cold spot area mainly distributes in the Northeastern Yunnan, Northwestern Yunnan and Southwestern Yunnan. This study could provide a scientific basis for SFQ management and ecological environment protection in the plateau mountain region.

**Keywords:** land evaluation; slope farmland quality; evaluation model; spatial autocorrelation; spatial structure; GIS; Yunnan Province

## 1. Introduction

Farmland is one of the most precious natural resources, as well as the basic resource to ensure food security and human sustainable development [1–3]. However, more and more farmland has been occupied in the world with the acceleration of urbanization in the past 60 years. Moreover, unreasonable cultivation results in the decrease in both the quantity and quality of farmland, thus seriously threatening food security and sustainable development of human beings [4–7]. Especially in some developing countries, the quantity and quality of farmland has decreased more significantly [8]. China is a typical developing country, and there are many problems such as farmland quality decrease, spatial fragmentation, ecological problems, etc. Hence, it has always been a topic of great concern

to scholars at home and abroad regarding how to protect the quantity and quality of farmland so as to meet the food demand of more than 1.3 billion people [9]. To this end, the Chinese government has implemented the three-in-one protection strategy of quantity, quality and ecology of farmland in order to comprehensively maintain and improve the production, ecology and living functions of the farmland ecosystem [10].

As an important part of farmland, slope farmland is a basic resource for agricultural production in hilly areas of the world [11,12]. Slope farmland generally refers to the dry farmland distributed on the hillside with poor flatness, serious soil and water loss as well as low crop yield. Simultaneously, the slope farmland is an important part of farmland resources in China, while additionally, their quantity and quality are closely related to food security and farmers' income in hilly and mountainous areas [13]. The proportion of slope farmland to land in the hilly and mountainous areas is greater in Central and Western China, and it is also an important basic resource for agricultural production [14]. As a typical plateau mountain region in China, the slope farmland in Yunnan has characteristics such as a wider distribution area, greater slope, serious water and soil loss, frequent seasonal drought, low soil quality, etc. [15]. Therefore, it is the premise and basis to scientifically evaluate the slope farmland quality (SFQ) in Yunnan Province and analyze the spatial structure features of SFQ to implement the three-in-one protection strategy of quantity, quality and ecology of farmland.

The concept of farmland quality is derived from soil quality. Soil quality refers to an ability to support soil and maintain crop growth. Good soil quality not only has a higher productivity but is also of great significance to improve the regional ecological soil and water environment [16]. With the increasingly serious problems in food security, global climate change, farmland degradation, and so on, the concerns people have in relation to the soil fertility, land suitability, potential productivity and ecological environment security gradually change, with a focus ranging from soil quality to farmland quality [17]. Although the definition of farmland quality has not been unified, the farmland quality connotation has gradually been understood. Generally, the "farmland quality" could reflect the production capacity, farmland environment and product quality of farmland, as well as a sum of soil quality, spatial geography quality, management quality and economic quality of farmland [18]. As such, the evaluation of farmland quality is to measure the state of farmland quality or the degree to which the farmland meets its function and demand. Moreover, reasonable evaluation of farmland quality can reflect the spatial differences, evolution rules and influencing factors of farmland quality, which can provide the scientific basis to protect the farmland quality [19,20].

Research on the evaluation of farmland quality is carried out mainly from the index system and model. In terms of the index system evaluation, the farmland system is a synthesis of natural and socio–economic factors, so its evaluation indexes generally include natural factors and socio–economic factors [21]. At present, the differentiated evaluation index system is established based on connotation analysis of farmland quality, demand analysis of farmland quality, limiting factor diagnosis, land consolidation and reconstruction of the human–earth relationship. Due to the difference of cognition and evaluation purpose of cultivated land quality, the index system proposed by different researchers is quite different, and the index classification and screening criteria are not unified [7,22]. From the existing research, soil fertility, climate and environment are usually taken as the main indexes to evaluate the farmland quality [23]. For example, Xia et al. [24] extracted farmland quality factors by a Ggaofen-one remote sensing satellite, thus, establishing the farmland quality evaluation index system based on the satellite image. Wang et al. [25] constructed the farmland quality evaluation index system on the basis of the three-in-one idea of quantity, quality and ecology. Compared with other farmland types, the slope farmland utilization is unique. However, until now, there is little research on the SFQ evaluation system.

In addition, for the farmland quality evaluation model, in 1961 the United States Department of Agriculture proposed the land capability classification (LCC) system. Since then, it has become a representative method to evaluate the land production potential [26]. Subsequently, a multi-objective farmland quality evaluation has been introduced, including land capacity monitoring, land production

potential classification, land bearing capacity evaluation, and so on. In 1976, Food and Agriculture Organization of the United Nations (FAO) proposed the first unified land evaluation system—A Framework for Land Evaluation—from a suitable perspective, moreover, its basic theory has become the foundation of many land productivity evaluation models [27]. Based on "A Framework for Land Evaluation", FAO further developed the agricultural ecological zone (AEZ) method to evaluate the potential of agricultural land use, and this model has been widely used in the world [28]. In the 1980s, the American Soil Conservation Service built the land evaluation and site assessment (LESA) system. This system includes two parts of land evaluation (LE) and site assessment (SA) [29,30].

At present, the frequently-used farmland quality evaluation model includes the weighted sum method, land productivity evaluation method, Pressure-state-response (P-S-R) frame model method, analytic hierarchy process, fuzzy comprehensive evaluation method, farmland potential evaluation method, suitability evaluation method, soil environment quality evaluation method, etc. Among them, the weighing-sum method, land productivity evaluation method, and P-S-R frame model method are the most widely used [31]. In addition, the introduction of GIS, remote sensing(RS), etc. has promoted the substantial development of research on the evaluation of farmland quality in regards to the updated data and evaluation accuracy [32–34]. In China, with the study of farmland quality evaluation and deep practice, the farmland quality evaluation standard system has gradually formed, represented by agriculture land classification, farmland productivity evaluation and farmland environment quality evaluation [35].

In recent decades, the farmland quality significantly degenerates, so the farmland quality evaluation has become a hot topic for research. The slope farmland is a special part of farmland, and the influencing factors of SFQ are different from those of other types of farmland. However, at present, there is little research on the SFQ evaluation. Hence, the research on the evaluation index system construction and evaluation model establishment of SFQ still need to be strengthened. Furthermore, the analysis of spatial structure characteristics of farmland quality is the foundation of farmland quality partition protection [36]. However, most of the existing research only describes the spatial and temporal distribution characteristics of farmland quality, and there is less research on the spatial variability and spatial clustering feature of farmland quality. As a typical plateau mountain region, the slope farmland is an important part of farmland resources in Yunnan. Therefore, it is of great significance to scientifically evaluate the SFQ for sustainable utilization and protection of regional slope farmland.

Here, this paper constructs the SFQ evaluation system based on the utilization characteristics and key influencing factors of the slope farmland in Yunnan Province. Thus, the SFQ in Yunnan is quantitatively evaluated and the spatial structure characteristics of SFQ are systematically analyzed, which can provide the scientific basis for the SFQ management and ecological environment protection of sloping farmland in the plateau mountain area.

## 2. Study area and Data Sources

### 2.1. Study Area

Yunnan is located in the Southwest border and Southwest Yunnan–Guizhou Plateau, with a land area of 0.39 million $km^2$, in which the East Asian monsoon and South Asian monsoon converge. For the topography, it is high in the north and low in the south. Moreover, the mountainous area accounts for 84% of the total land area, and the hill accounts for only 10%. The topography gradually develops from mountain land to karst landform. Due to the common influence of ecological environment evolution and human activities, the region with moderate ecological vulnerability accounts for 32.02% of the total area, while the region with strong and extremely strong fragility accounts for 53.63% [37]. There is abundant rainfall and many rivers in Yunnan, but they are unevenly distributed in space and time. The average reference crop evaporation is between 786.3 and 1511.6 mm, with a mean of 1090.4 mm [38].

The main soil types include red earths, lateritic red earths, purplish soils, yellow earths and yellow–brown earths. Most of the farmland in the region has a slope of more than 3°, so the agricultural production of slope farmland plays an important role in the agricultural activities in Yunnan. The research results show that the area of slope farmland in Yunnan is 4.7255 million hm², accounting for 69.79% of the farmland area. The agricultural production of slope farmland plays an important role in the agricultural activities in Yunnan [15]. In order to make the SFQ evaluation consistent with the regional agricultural activities, Yunnan is divided into seven regions in this study, according to the comprehensive agricultural partition in Yunnan. The elevation and spatial distribution of slope farmland are shown in Figure 1, and the distribution characteristics of slope farmland in different partitions are listed in Table 1.

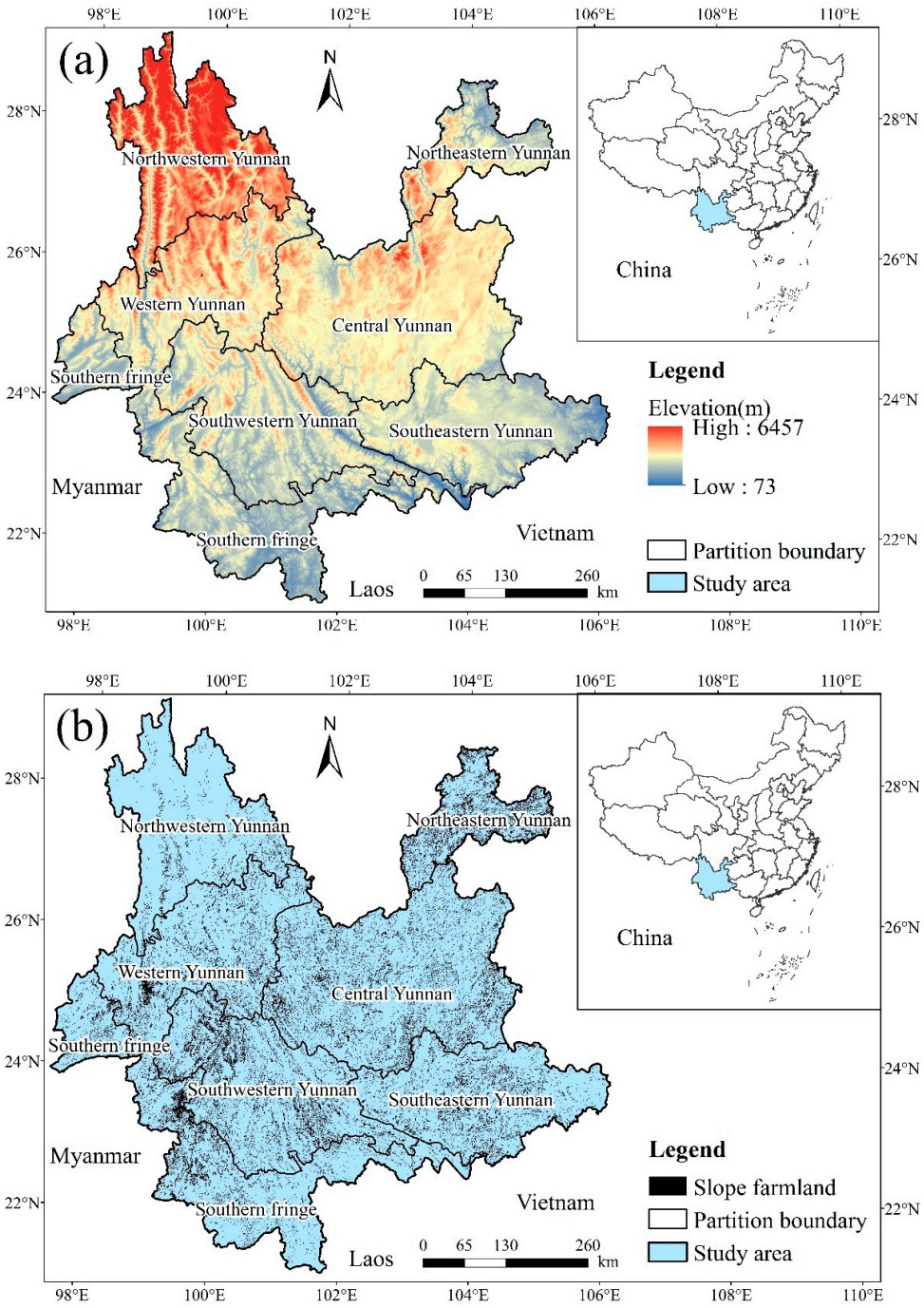

**Figure 1.** Elevation (**a**) and slope farmland (**b**) spatial distribution in the study area.

**Table 1.** Distribution characteristics of slope farmland in different partitions of Yunnan Province.

| Partition Name | Arable Land | | | Dry Land | | | Slope Farmland | | |
|---|---|---|---|---|---|---|---|---|---|
| | Area ($\times 10^4 hm^2$) | Proportion of Land Area | Average Slope (°) | Area ($\times 10^4 hm^2$) | Average slope (°) | Proportion of Cultivated Land Area | Area ($\times 10^4 hm^2$) | Average Slope (°) | Proportion of Cultivated Land Area |
| Central Yunnan | 187.76 | 21.32% | 9.70 | 132.16 | 11.69 | 70.39% | 114.80 | 13.20 | 61.14% |
| Western Yunnan | 91.94 | 18.13% | 12.28 | 62.71 | 15.02 | 68.21% | 58.10 | 16.09 | 63.20% |
| Southeastern Yunnan | 95.73 | 19.02% | 10.90 | 67.06 | 11.97 | 70.05% | 58.55 | 13.46 | 61.16% |
| Southwestern Yunnan | 109.20 | 18.46% | 16.31 | 96.15 | 17.00 | 88.05% | 93.60 | 17.41 | 85.71% |
| Southern Fringe | 99.03 | 15.90% | 12.61 | 77.89 | 14.39 | 78.66% | 71.32 | 15.58 | 72.03% |
| Northeastern Yunnan | 62.95 | 27.61% | 16.08 | 55.68 | 16.45 | 88.46% | 52.80 | 17.25 | 83.88% |
| Northwestern Yunnan | 30.40 | 5.95% | 18.01 | 25.74 | 19.14 | 84.66% | 23.34 | 20.97 | 76.77% |
| Total | 676.99 | 17.61% | 12.68 | 517.39 | 14.41 | 76.42% | 472.55 | 15.62 | 69.79% |

## 2.2. Data Sources

The digital elevation model (DEM) data of research region come from the Geospatial Data Cloud Platform of Chinese Academy of Sciences (http://www.gscloud.cn) with a spatial resolution of 30 m. This data set is obtained by processing the data of the first edition of Advanced Spaceborne Thermal Emission and Reflection Radiometer Global Digital Elevation Model (ASTER GDEM), and it is a digital elevation data product with a spatial resolution of 30 m in the world. The land use data in Yunnan are from the Resource and Environment data Cloud Platform of Chinese Academy of Sciences (http://www.resdc.cn), with a spatial resolution of 30m. The basic data, such as thickness of the cultivated-layer, pH value of the soil, organic matter, total nitrogen, available phosphorus, rapidly available potassium, probability of irrigation, etc., are from the data set obtained from the formula fertilization by soil testing in Yunnan Province (2015). The basic data of effective soil thickness, soil bulk density and soil texture are from the data set of main properties of farmland quality in southern China published by the Ministry of Agriculture and Rural Affairs of China (2018).

## 3. Methodology

### 3.1. Quality Evaluation System of Slope Farmland

#### 3.1.1. Evaluation Unit

The evaluation unit is the basic unit of SFQ evaluation. In this study, the evaluation scale is a provincial region. According to the previous findings [39,40], the 30 m × 30 m high-precision grid (pixel) is taken as the minimum unit of the SFQ evaluation. Each grid (pixel) has the same attributes such as land use, soil type, elevation, and so on. This partition method of the evaluation unit can be conveniently used for the classifying and summarizing according to the standards, i.e., different patch units, agricultural partition, soil types, district ranges, and so on, with less loss of summary result information.

#### 3.1.2. Construction of Evaluation Index System

The slope farmland system is a compound system formed by multiple factors such as climate, soil, geomorphology and human cultivation activities with the interaction. Moreover, its quality is affected by multiple factors [21]. As we select the SFQ evaluation index, the influence of each factor on the SFQ should be taken into account, so as to select the main factors as the evaluation indexes. In terms of the connotation, the slope farmland not only has the general attributes of farmland but also has its own special attributes, that is, larger soil erosion intensity, susceptibility to drought, obvious nutrient degradation, spatial distribution fragmentation, etc. [41,42].

Therefore, both the basic indexes of farmland and the characteristic indexes reflecting the slope farmland utilization should be considered for the selection of the SFQ evaluation index. Then, the primary index system is determined. Based on this, the evaluation indexes are discriminated and screened in order to eliminate the information overlap caused by the interaction among the indexes with a statistical analysis method, thus establishing the index system of minimum data set (MDS) [43,44]. Based on the above evaluation index selection principle and analytic hierarchy process, the index system of minimum data set for SFQ evaluation in Yunnan Province is established, as shown in Table 2. The criterion layer of the evaluation index system includes soil profile properties (B1), physicochemical properties (B2), soil nutrient (B3), site conditions (B4), spatial form (B5), moisture condition (B6) and soil erosion (B7), a total of seven dimensions. Furthermore, each criterion layer contains many specific indexes.

**Table 2.** Index system of slope farmland quality evaluation in Yunnan Province.

| Target Layer (A) | Criterion Level (B) | | Index Level (C) | |
|---|---|---|---|---|
| | Code | Classification | Code | Index |
| Slope farmland quality (A) | B1 | Soil profile properties | C1 | Effective soil layer thickness (cm) |
| | | | C2 | Thickness of cultivated-layer (cm) |
| | B2 | Physicochemical properties | C3 | Bulk density (g/cm$^3$) |
| | | | C4 | Soil texture (Dimensionless) |
| | | | C5 | pH value (Dimensionless) |
| | | | C6 | Organic matter (g/kg) |
| | B3 | soil nutrient | C7 | Available phosphorus (mg/kg) |
| | | | C8 | Available potassium (mg/kg) |
| | B4 | Site conditions | C9 | ≥10 °C accumulated temperature (°C) |
| | B5 | Spatial form | C10 | Field regularity (Dimensionless) |
| | | | C11 | Degree of continuity (Dimensionless) |
| | B6 | Moisture condition | C12 | Rainfall (mm) |
| | | | C13 | Irrigation assurance rate (%) |
| | B7 | Soil erosion | C14 | Slope (°) |

### 3.1.3. Quantification of Evaluation Index

In the evaluation indexes listed in Table 2, the indexes belong to the quantitative index in addition to the cultivated-layer texture of the conceptual index. According to the Chinese national standard Grade of Cultivated Land Quality (GB/T33469-2016), the membership of soil texture is assigned to be translated into the quantifiable evaluation index. For the quantitative indexes, the available soil thickness, cultivated-layer thickness, soil bulk density, pH value, organic matter, total nitrogen, available phosphorus, available potassium, elevation, accumulated temperature greater than or equal to 10 °C, rainfall, irrigation guarantee rate and farmland slope can be directly measured or obtained by investigation. However, the slope farmland quality indexes of regularity and continuity degree are obtained by analysis and calculation. The spatial morphology evaluation of slope farmland can be quantitatively measured from the geometrical shape of the farmland and the spatial continuity. Among them, the field regularity is described by the fractal dimension of landscape ecology (*FRAC*) [45] with a value range of 1.0~2.0. Here, the smaller the index is, the more regular the field shape is. Otherwise, the field shape is more complex.

Furthermore, the intensive large-scale utilization degree of slope farmland can be expressed by the continuity degree *Q* of slope farmland, and its value can be quantified according to the field size. Moreover, the larger the *Q* value is, the higher the continuity degree is. Otherwise, the continuity degree is lower [46].

The *FRAC* and *Q* are calculated by the following formulas.

$$FRAC = \frac{2\log(P/4)}{\log(a)} \tag{1}$$

$$Q = \begin{cases} 20 \ (a \leq 2.5 \text{ hm}^2) \\ 20 + 80\dfrac{a - 2.5}{78 - 2.5} \ (2.5 < a \leq 78 \text{ hm}^2) \\ 100 \ (a > 78 \text{ hm}^2) \end{cases} \tag{2}$$

where *FRAC* represents the regularity degree of slope farmland. *Q* is the continuity degree of slope farmland. *P* is the perimeter of slope farmland, m. *a* is the slope farmland area, m$^2$. In addition, the range of *FRAC* value is between 1 and 2, and the range of *Q* value is between 20 and 100. The area threshold of slope farmland is obtained by the natural break point method. Moreover, the minimum threshold is 2.5 hm$^2$, and the maximum threshold is 78 hm$^2$.

### 3.1.4. Membership Function

In addition to the assignment for the soil texture membership of cultivated-layer according to the Chinese national standard Cultivated Land Quality Grade (GB/T33469-2016), other indexes select the membership function according to their positive and negative effects on the SFQ. According to the effects of the evaluation index and SFQ, the membership function can be divided into three types, namely, S-type membership function, anti-S-type membership function and parabolic-type membership function.

Correspondingly, some indexes, such as the available soil layer thickness, cultivated-layer thickness, organic matter, total nitrogen, available phosphorus, available potassium, accumulated temperature greater than or equal to 10 °C, rainfall and irrigation guarantee rate, etc., have a positive effect on the SFQ, and they select the S-type membership function. However, some indexes have a negative effect on the SFQ, such as bulk density of soil, elevation, farmland regularity, farmland slope, etc., so they select the anti-S-type membership function. The pH value of soil has a peak effect on the SFQ, thus selecting the parabolic-type membership function. According to the change range of SFQ evaluation index value of Yunnan Province, the parameter value of each evaluation index membership function is comprehensively determined by referring to relevant research results, as shown in Tables 3 and 4.

**Table 3.** S-type and anti-S-type membership functions and parameters.

| Index Code | Index | Membership Function Type | Formula of Membership Function | Parameters *a* | Parameters *b* |
|---|---|---|---|---|---|
| C1 | Effective soil layer | S-type | | 30.0 | 120.0 |
| C2 | Thickness of cultivated-layer | S-type | | 14.03 | 20.00 |
| C6 | Organic matter | S-type | | 15.0 | 40.0 |
| C7 | Available phosphorus | S-type | $\mu(x) = \begin{cases} 1, x \geq b \\ 0.9(x-a)/(b-a)+0.1, a<x<b \\ 0.1, x \leq a \end{cases}$ | 5.70 | 53.83 |
| C8 | Available potassium | S-type | | 40.00 | 216.06 |
| C9 | ≥10 °C accumulated temperature | S-type | | 3000.0 | 5500.0 |
| C11 | Degree of continuity | S-type | | 20.0 | 100.0 |
| C12 | Rainfall | S-type | | 800.0 | 1200.00 |
| C13 | Irrigation assurance rate | S-type | | 40.0 | 100.0 |
| C3 | Soil bulk density | Anti-S-type | | 1.15 | 1.50 |
| C10 | Field regularity | Anti-S-type | $\mu(x) = \begin{cases} 1, x \leq a \\ 0.9(x-b)/(a-b)+0.1, a<x<b \\ 0.1, x \geq b \end{cases}$ | 1.00 | 1.34 |
| C14 | Slope | Anti-S-type | | 3.00 | 25.00 |

Note: $\mu(x)$ is a membership function, *x* is the measured value of evaluation index, *a* and *b* are the lower limit and upper limit of the critical value, respectively, which are determined according to the measured results of the study area and the comprehensive comparison of relevant domestic research results.

**Table 4.** Parabolic membership functions and parameters.

| Index Code | Index | Membership Function Type | Formula of Membership Function | Parameters | | | |
|---|---|---|---|---|---|---|---|
| | | | | $a_1$ | $b_1$ | $b_2$ | $a_2$ |
| C5 | pH value | Parabolic-type (Peak type) | $\mu(x) = \begin{cases} 1, b_2 \geq x \geq b_1 \\ 0.9(x - a_1)/(b_1 - a_1) + 0.1, a_1 < x < b_1 \\ 0.9(x - a_2)/(b_2 - a_2) + 0.1, a_2 > x > b_2 \\ 0.1, x \leq a_1 \, \text{or} \, x \geq a_2 \end{cases}$ | 5.5 | 6.5 | 7.5 | 8.5 |

Note: $\mu(x)$ is the membership function, $x$ is the measured value of the evaluation index, $a_1$ and $b_2$ are the lower limit and upper limit of the critical value of the index, respectively, the minimum value and the maximum value of the measured value are taken in this study; $b_1$ and $b_2$ are the upper and lower boundary points of the most suitable value, and their values are determined according to the comprehensive comparison of the measured results in the study area.

### 3.1.5. Index Weight

In order to improve the accuracy of weight calculation, the evaluation index weight is determined by combining the principal component analysis (PCA), analytic hierarchy process (AHP) and entropy weight method (EWM). During the calculation process, the PCA, AHP and EWM are used to calculate the weight of each index. Simultaneously, the comprehensive weight is taken as the final weight of the evaluation, and the comprehensive weight is calculated as follows.

$$C_i = \frac{C_{i1} \times C_{i2} \times C_{i3}}{\sum_{i=1}^{n} C_{i1} \times C_{i2} \times C_{i3}} \tag{3}$$

where $C_i$ is the comprehensive weight of the $i$-th evaluation index. $C_{i1}$ is the weight obtained by the calculation of PCA. $C_{i2}$ is the weight obtained by the calculation of AHP. And $C_{i3}$ is the weight obtained by the calculation of EWM.

### 3.1.6. Evaluation Model and Quality Grading

Sloping integrated fertility index (SIFI) was calculated by the weighted sum method.

$$\text{SIFI} = \sum_{i=1}^{n} (K_i \times C_i) \tag{4}$$

where SIFI is the SFQ index (dimensionless), and the numerical value range is between 0 and 1. $K_i$ is the membership value of the $i$-th evaluation index. $C_i$ is the weight of evaluation index of the $i$th evaluation index. $n$ is the number of evaluation indexes.

According to the Cultivated Land Quality Grade, the farmland quality is divided into ten grades with the equidistance method in sequence from small to large between the highest point and the lowest point of the comprehensive farmland quality index curve. It is found that the larger the farmland quality index is, the higher the farmland quality level is. Thus, the quality of the first-class farmland is the highest, and the quality of ten-class farmland is the lowest.

### 3.2. Research Methods of Spatial Structure Characteristics of SFQ

#### 3.2.1. Geostatistical Analysis Method

As a spatial variable, the spatial change feature of SFQ is structural and random. The spatial variability is measured with the semi-variance function [47,48]. The semi-variance function is calculated as follows.

$$r(h) = \frac{1}{2n(h)} \sum_{i=1}^{n(h)} [Z(x_i) - Z(x_i - h)]^2 \tag{5}$$

where $r(h)$ is the semi-variance variation function, and it can characterize the spatial variation pattern on the whole scale. $h$ is the spatial segmentation distance between two sample points. $Z(x_i)$ and $Z(x_i+h)$ are the values of spatial positions $x_i$ and $x_i+h$, respectively. $n(h)$ is the logarithm of sample with a spatial distance of $h$.

The spatial variability of the research object is distinguished according to the parameters of the variability function. Among them, the size of nugget $C_0$ can reflect the influence degree of randomness factors on the regional SFQ. Furthermore, the Sill ($C_0+C$) represents the maximum spatial variation degree of SFQ. The spatial distribution of SFQ indexes is moderately autocorrelated. Nugget coefficient $C_0/(C_0+C)$ represents the proportion of random variation to the total variation of slope farmland. If the proportion is less than 25%, the spatial distribution of SFQ indexes belongs to the strong spatial autocorrelation. If the proportion is between 25% and 75%, the spatial distribution of SFQ indexes belongs to moderate spatial autocorrelation. If the proportion is greater than 75%, the spatial distribution of SFQ indexes belongs to a weak spatial correlation, indicating that the degree of spatial heterogeneity caused by the random part plays a major role [49,50].

3.2.2. Global Spatial Autocorrelation Analysis Method

Spatial autocorrelation indices were used to quantify landscape patterns. The global spatial autocorrelation of SFQ grades can be used to study the aggregation or dispersion degree of SFQ on the whole. This paper uses the global Moran's *I* index to calculate the spatial autocorrelation characteristics of SFQ grades [51,52], and the calculation formula is as follows.

$$I = \frac{N}{\sum_{i=1}^{N} \sum_{j=1}^{N} W(i,j)} \cdot \frac{\sum_{i=1}^{N} \sum_{j=1}^{N} W(i,j) \cdot (X_i - \overline{X}) \cdot (X_j - \overline{X})}{\sum_{i=1}^{N} (X_i - \overline{X})^2} \tag{6}$$

where $N$ is the number of research objects. $X_i$ is the observed value, and $\overline{X}$ is the mean of $X_i$. $W(i,j)$ is the spatial connection matrix between the research objects $i$ and $j$.

Moreover, the Moran's *I* value of global spatial autocorrelation is between −1 and 1. If Moran's *I* value is more than 0, the space is positively correlated, as well as the research object presents the spatial aggregation characteristic. However, if the Moran's *I* value is less than 0, the space is negatively correlated, as well as the research object presenting a random distribution. Furthermore, *P* value is used for the significance test.

3.2.3. Local Spatial Autocorrelation Analysis Method

The global Moran's I index can test the overall spatial distribution pattern of SFQ grade rather than reflect the spatial correlation and local significance level of elements or attributes between adjacent evaluation units [50,53]. This paper uses the local Moran's *I* index to analyze the spatial correlation characteristics of local SFQ grade. Then, the calculation formula of local Moran's *I* is as follows.

$$I_i = \frac{X_i - \overline{X}}{\sqrt{\sum_{i=1}^{n}(X_i - \overline{X})/(n-1)}} \cdot \sum_{j=1}^{N} W(i,j)(X_j - \overline{X}) \tag{7}$$

where $N$ is the number of research objects. $X_i$ is the observed value, and $\overline{X}$ is the mean of $X_i$. $W(i,j)$ is the spatial connection matrix between the research objects $i$ and $j$.

Moreover, the Moran's *I* value of global spatial autocorrelation is between −1 and 1. If Moran's *I* value is more than 0, the space is positively correlated, as well as the research object presenting the spatial aggregation characteristic. However, if the Moran's *I* value is less than 0, the space is negatively correlated, as well as the research object presenting the spatially discrete characteristic. If Moran's *I* value is 0, the research object randomly distributes. Furthermore, *p* value is used for the significance test.

### 3.2.4. Spatial Cold and Hot Spots Analysis Method

The cold and hot spots analysis is one of the commonly-used methods to explore the local spatial clustering distribution characteristics. This method could collect the Getis-Orid $G_i^*$ index of each element by calculating the elements, so as to determine whether there is high-value clustering or low-value clustering as well as to identify its location [50,54]. Furthermore, Getis-Orid $G_i^*$ is a commonly-used index to describe the regional cold and hot spots, thus reflecting the correlation degree of SFQ grade between different research areas. The significance degree of $Z$ $(G_i^*)$ is used to identify the spatial distribution of hot and cold spots in different regions.

In this paper, the natural discontinuity point method is used for $Z(G_i^*)$ classification. Simultaneously, the spatial cold and hot areas under different confidence levels are divided according to three confidence coefficients of 99%, 95% and 90%. The computational formulas of $G_i^*$ and $Z$ $(G_i^*)$ are as follows.

$$G_i^*(d) = \sum_{j=1}^{n} W_{ij}(d)X / \sum_{j=1}^{n} X_{ij} \tag{8}$$

$$Z(G_i^*) = (G_i^* - E(G_i^*)) / \sqrt{Var(G_i^*)} \tag{9}$$

### 3.3. Data Calculation and Analysis

In this study, Excel 2013 software is used for basic data processing, likewise, SPSS19.0 software for data statistical analysis, ArcGIS 10.2 software for spatial data analysis and calculation. The spatial autocorrelation analysis was carried out by using Geoda software, and a Moran scatter plot and local indicators of spatial association (LISA) map were drawn to realize the visual expression of spatial autocorrelation features. In spatial autocorrelation analysis, the spatial weight should be determined first. There are two methods to determine the spatial weight matrix, which are based on adjacency relationship and distance relationship. Referring to the existing research results, this paper uses the K-nearest method to determine the spatial weight [55].

## 4. Results and Discussion

### 4.1. Spatial Distribution Characteristics of SFQ

Figure 2a shows the spatial distribution of SFQ indexes. It can be seen that SIFI of Yunnan distributes between 0.36 and 0.81, with an average value of 0.59 ± 0.06. Moreover, the SIFI of most evaluation units is less than 0.6, and indeed the SIFI is generally lower. The difference in the SIFIs of different evaluation units is significant ($p < 0.05$), which can reflect the spatial difference of SFQ in different regions. From the spatial distribution, the SFQ indexes present a staggered coupling distribution characteristic of spatial aggregation and variability. Among them, the SFQ indexes are higher in the central Yunnan, the Southern Fringe, Western Yunnan, Southeastern Yunnan, while they are lower in Southwestern Yunnan and Northeastern Yunnan. In addition, there exists a certain difference in the means of SFQ indexes in different regions, but the difference is relatively smaller. The SFQ indexes in the Southern Fringe and Western Yunnan are relatively larger, while they are relatively smaller in the Northwestern Yunnan and Northeastern Yunnan.

Figure 2b shows the spatial distribution of SFQ grades in Yunnan. Similarly, there exists a difference in the distribution area of SFQ grades in different regions, and the spatial distribution of SFQ grades displays obvious aggregation characteristics. The SFQ is based on sixth-class farmland, fifth-class farmland, seventh-class farmland and fourth-class farmland. And other grades of slope farmland distribute less, so the quality grade is generally lower. From the distribution of SFQ grades in different regions, the SFQ grade is relatively higher in the Southern Fringe, Central Yunnan, Western Yunnan and Southeastern Yunnan, mainly based on fourth-class farmland and fifth-class farmland.

However, the SFQ grade is relatively lower in Northeastern Yunnan and Northwestern Yunnan, mainly based on sixth-class farmland and seventh-class farmland.

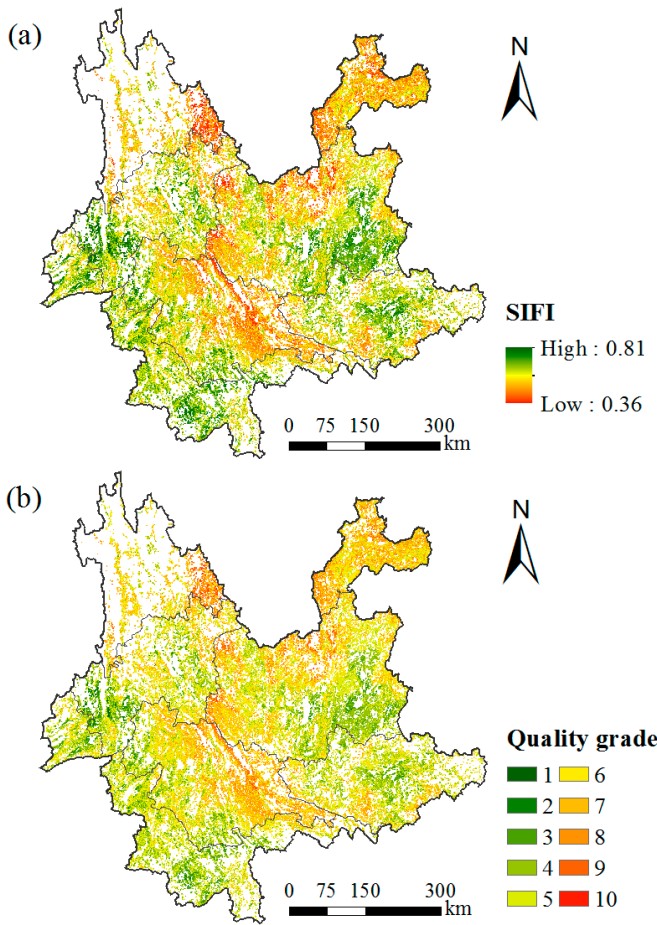

**Figure 2.** Spatial distribution of sloping integrated fertility index (SIFI) (**a**) and quality grade (**b**) of slope farmland.

The quality of farmland is closely related to regional food security and ecological security, so special protection policies are generally implemented for farmland in the world. China implements the strictest cultivated land protection system in the world, but the pure administrative restraint mechanism cannot change the declining trend of farmland quantity and quality [56]. Studies have shown that the quality of farmland in China is generally low, and the medium and low-grade farmland accounts for more than two thirds of the total farmland area [57]. Due to the differences of nature and cultivation conditions in different regions of China, the quality of farmland varies greatly in different regions. Wei et al. show that the overall quality of farmland in Shaanxi Province is relatively low, and it shows a downward trend in the recent 10 years [58]. Wang et al. shows that the overall quality of farmland in Shandong Province is relatively high, but it has a slight downward trend [34]. Yang et al. also showed that the quality of farmland in Yunnan Province is generally at a medium–low level [59]. Although the farmland quality evaluation models used in the above study are different, and the results cannot be directly compared, but they can reflect the characteristics of farmland quality in different regions. As an important part of farmland, slope farmland has the characteristics of soil erosion, drought and water shortage compared with other farmland types. Our research shows that most of the SFQ grade in Yunnan is in the sixth, fifth and seventh grade, and the SFQ is generally low. This conclusion is consistent with the actual situation of slope farmland utilization in Yunnan Province.

Table 5 shows the areas and distribution proportions of different SFQ grades. It can be seen that the distribution areas of different SFQ grades present a pattern of a larger middle and both have

smaller ends. Among the ten SFQ grades, the sixth-class farmland distributes most widely with an area of 1.2163 million hm$^2$, accounting for 25.74% of the total area of slope farmland. The second one is the fifth-class farmland with a distribution area of 1.1801 million hm$^2$, accounting for 24.97% of the total area of slope farmland. Likewise, the distribution areas of fourth-class and seventh-class farmland are also larger, with 0.7975 million hm$^2$ and 0.8594 million hm$^2$, respectively, accounting for 16.88% and 18.19% of the total area of slope farmland, respectively. However, the distribution areas of fourth-class and seventh-class farmland are smaller, with the distribution areas of 0.2477 million hm$^2$ and 0.3306 million hm$^2$, respectively. Similarly, the distribution areas of first-class, second-class farmland, ninth-class farmland and tenth-farmland have the smaller distribution area, all accounting for less than 2%.

**Table 5.** Distribution area statistics of slope farmland quality (SFQ) grades in different regions of Yunnan Province.

| Quality Grade | Area Proportion of Different Zones (%) | | | | | | | Total | |
| | Central Yunnan | Western Yunnan | Southeastern Yunnan | Southwestern Yunnan | Southern Fringe | Northeastern Yunnan | Northwestern Yunnan | Area (×10$^4$hm$^2$) | Proportion (%) |
|---|---|---|---|---|---|---|---|---|---|
| 1 | 0.00 | 0.72 | 0.00 | 0.00 | 0.00 | 0.00 | 0.00 | 0.45 | 0.09 |
| 2 | 0.18 | 3.74 | 0.01 | 0.11 | 0.60 | 0.00 | 0.00 | 3.07 | 0.65 |
| 3 | 5.05 | 11.98 | 4.74 | 2.59 | 8.78 | 0.00 | 0.03 | 24.77 | 5.24 |
| 4 | 19.34 | 27.06 | 22.86 | 9.18 | 25.41 | 0.09 | 2.00 | 79.75 | 16.88 |
| 5 | 23.35 | 24.88 | 36.03 | 20.16 | 46.01 | 1.97 | 7.96 | 118.01 | 24.97 |
| 6 | 26.82 | 21.76 | 27.96 | 28.83 | 16.67 | 29.49 | 33.72 | 121.63 | 25.74 |
| 7 | 17.22 | 8.47 | 6.76 | 25.46 | 2.42 | 50.64 | 28.39 | 85.94 | 18.19 |
| 8 | 6.49 | 1.33 | 1.61 | 12.35 | 0.10 | 16.13 | 19.17 | 33.06 | 7.00 |
| 9 | 1.49 | 0.08 | 0.03 | 1.30 | 0.00 | 1.68 | 7.90 | 5.61 | 1.19 |
| 10 | 0.06 | 0.00 | 0.00 | 0.02 | 0.00 | 0.00 | 0.83 | 0.27 | 0.06 |

## *4.2. Spatial Variation Characteristics of SFQ*

The geostatistical analysis model is a common method for the variability analysis of spatial variables. The geostatistical analysis is suitable for the sample data with normal distribution. Hence, the data distribution type should be firstly tested before the geostatistical analysis. Based on the exploratory analysis function for data, the ArcGIS10.2 software is used to plot a histogram of sample distribution, cumulative distribution curve, and normal Q-Q plot to analyze the spatial distribution type of SFQ indexes. Figure 3 shows the exploratory analysis results of spatial distribution of SFQ index. As can be seen from Figure 3a, the frequency histogram of SFQ indexes in Yunnan Province presents a typical normal distribution characteristic of frequency. It is found from the cumulative curve change in Figure 3b that the cumulative curve of sample frequency is S-shaped, which suggests that the sample distribution has typical normal distribution characteristic. From the normal Q-Q Plot in Figure 3c, the sample data concentratedly distribute in the line characterizing the normal distribution. The above analysis indicates that the spatial distribution of SFQ indexes in Yunnan presents a better normal distribution characteristic. This is consistent with the conclusion of research on distribution characteristics of farmland quality grade in Yunnan by Zeng et al. [60].

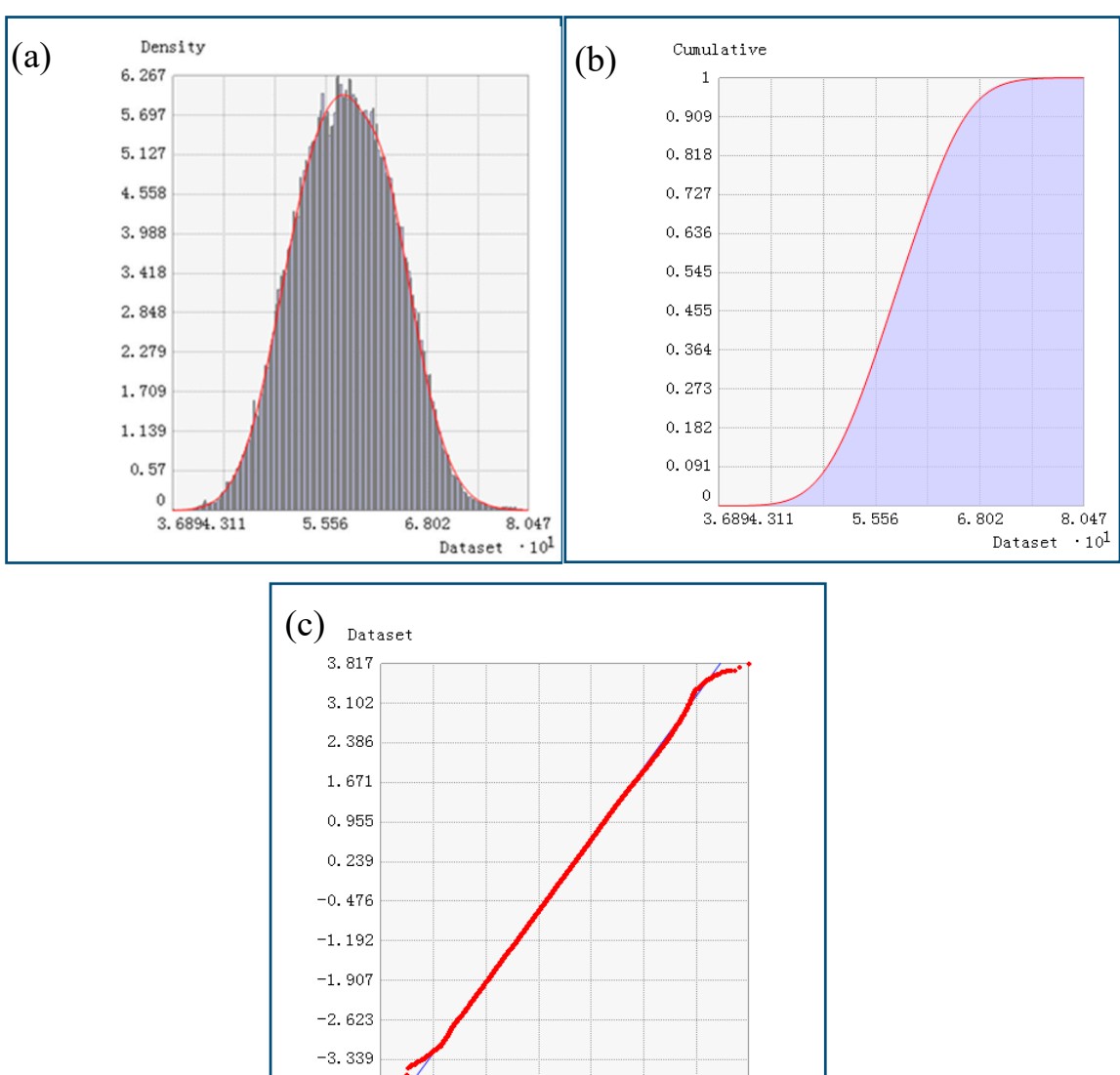

**Figure 3.** Exploratory analysis results of spatial distribution of SFQ index. (**a**) Density histogram, (**b**) Cumulative curve, (**c**) Normal Q-Q plot.

The selection of variable function is an important part of spatial variability analysis. In order to obtain the optimal variation function of the SFQ index distribution, four kinds of theoretical models, namely, the Circular model, Spherical model, Exponential model, and Gaussian model are used to fit the variation function. Then, the optimal model is selected by contrasting some parameters such as mean, mean square root, standardized mean, Root–Mean–Square Standardized, mean standard error. Among them, as the Root–Mean–Square Standardized is close to 1, the mean standard error is close to 0. Moreover, the smaller the other parameters are, the better the model fitting effect is [49].

Table 6 lists the fitting parameters of four theoretical models. Obviously, the four models can better fit the semi-variance function of SFQ indexes, and the difference among different fitting models is small on the whole. In addition, the Root–Mean–Square Standardized of Gaussian model is the closest to 1, and its mean standard error is the closest to 0. Therefore, it is suggested that the fitting effect of the Gaussian model is the best in the four models. In this paper, the Gaussian model is used to fit the semi-variance function of SFQ indexes.

**Table 6.** Summary of characteristic parameters of semi-variance function theory fitting model.

| Theoretical Model | Error Prediction Parameters | | | | |
|---|---|---|---|---|---|
| | Mean | Mean Square Root | Standardized Mean | Root-Mean-Square Standardized | Mean Standard Error |
| Circular model | −0.000158 | 0.0362 | −0.00323 | 0.7959 | 0.0455 |
| Spherical model | −0.000151 | 0.0362 | −0.00308 | 0.7953 | 0.0456 |
| Exponential model | −0.000105 | 0.036 | −0.0021 | 0.7889 | 0.0457 |
| Gaussian model | −0.000219 | 0.0364 | −0.00452 | 0.8038 | 0.0454 |

The geostatistical analysis method has been widely used in studying the spatial variation of soil quality elements and measuring land ecological space, but it is seldom used in the study of spatial pattern of farmland quality. Spatial heterogeneity of SFQ indexes is affected by both structural and random factors (also called non-structural factors) [50]. Among them, the random factors (external factors) include various obstacle factors and anthropogenic farming activities, while the structural factors (internal factors) include the factors dominating the SFQ, such as climatic condition, soil property, moisture condition, spatial morphological characteristic and so on.

According to the analysis results of spatial variability parameters of SFQ indexes, the Nugget $C_0$ is 0.5293, indicating that the random factors of spatial distribution of SFQ indexes are larger. The Partial Sill C is 0.4515, and the Sill ($C_0$+C) is 0.9808. Thus, the Nugget coefficient $C_0$/($C_0$+C) is 53.97%. This suggests that although the spatial variation caused by random factors is higher, the SFQ indexes are in the medium spatial autocorrelation on the whole. The structural factors, such as climatic condition, soil property, moisture condition, spatial morphological characteristic, etc., still play a major role. Furthermore, the influence of random factors on the spatial differentiation of SFQ indexes should not be ignored. This conforms to the great influence of unreasonable cultivation mode, soil erosion aggravation and frequent regional droughts of slope farmland in Yunnan in recent years. The range ($A_0$) refers to the distance as the variation function reaches the Sill as well as reflects the distance range of spatial autocorrelation of the research object. Here, the $A_0$ value of SFQ index in Yunnan is 17.96 km, which shows that the spatial autocorrelation range of SFQ in Yunnan is within 17.96km.

*4.3. Spatial Autocorrelation Characteristics of SFQ*

The autocorrelation characteristics among the spatial factors are mainly determined by global spatial autocorrelation and local spatial autocorrelation indexes. Here, the global spatial autocorrelation reflects the regional global spatial autocorrelation, while the local spatial autocorrelation further measures the correlation between the attribute eigenvalues of each geographic unit and those of its adjacent spatial units [51,52].

4.3.1. Global Spatial Autocorrelation Analysis

The SFQ grade is taken as the spatial attribute value, and the K-nearest method is used to determine the spatial weight. Then, the global Moran's *I* is calculated. According to the calculation results, Moran's *I* of the SFQ grade is 0.8489, Z is 1205.07, and *p* is 0.0000. As can be seen from the calculation results, Moran's *I* is more than 0, *p* is less than 0.01, and Moran's *I* is closer to 1. This indicates that the spatial distribution of the SFQ grade has a strongly positive spatial correlation, and the SFQ grade presents a significant spatial aggregation characteristic.

A Moran scatter plot can be used to qualitatively distinguish the relationship between some attribute values of a certain region and those of the surrounding region. The Moran scatter plot can be divided into four quadrants. That is, the first quadrant represents high–high (HH) aggregation, the second quadrant represents low–high (LH) aggregation, the third quadrant represents low–low aggregation (LL), and the fourth quadrant represents high–low (HL) aggregation. Figure 4 displays the Moran scatter plot of SFQ grades in Yunnan. Clearly, most of the SFQ grades focus on the first quadrant and the third quadrant, a few SFQ grades distribute in the second quadrant and the fourth

quadrant. The results show that the spatial autocorrelation type of SFQ is based on HH aggregation and LL aggregation, and the spatial distribution of SFQ grades present a higher spatial aggregation characteristic on the whole.

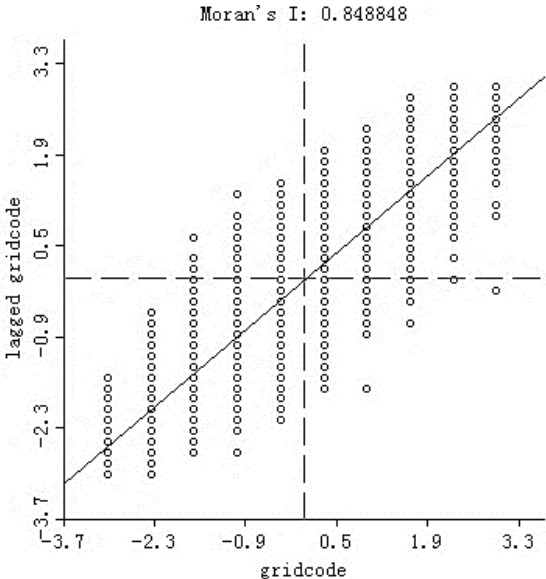

**Figure 4.** Moran scatter plot of SFQ grades in Yunnan.

Spatial autocorrelation analysis is a common method to explore the aggregation characteristics of spatial objects. In recent years, GIS based study on the spatial aggregation characteristics of farmland quality and protection zoning has been widely concerned. Wei et al. showed that the spatial distribution of farmland quality in Guangning County of China showed a certain aggregation rule, and the positive correlation types mostly appeared in the form of "group", while the negative correlation type had no obvious concentration area [61]. Li et al. showed that the farmland quality index in Zhejiang Province showed significant positive spatial correlation under different scales, and the global autocorrelation index of provincial spatial scale farmland quality index exceeded 0.8 [62]. The results of Yang et al. showed that the overall quality of cultivated land in Jiaozuo of Henan Province, was better, and showed a strong positive correlation in spatial distribution [55]. The research of Xiong et al. shows that there is a certain spatial positive autocorrelation in the distribution of village level cultivated land [63]. The results of this study are basically consistent with those of different regions mentioned above.

The global spatial autocorrelation index of each partition is calculated to further identify the global spatial autocorrelation characteristics of different SFQ grades in different regions. Table 7 lists global autocorrelation analysis results of SFQ grades in different regions. It is found that Moran's *I* value of SFQ grade in different regions is larger, all more than 0.6, moreover, they pass the significance test. This result suggests that the spatial distribution of SFQ in seven regions shows strong spatial positive correlation, and the SFQ grade has significant spatial aggregation characteristics.

**Table 7.** Summary of global autocorrelation analysis of SFQ grades in different regions.

| Inspection Parameters | Central Yunnan | Western Yunnan | Southeastern Yunnan | Southwestern Yunnan | Southern Fringe | Northeastern Yunnan | Northwestern Yunnan |
|---|---|---|---|---|---|---|---|
| Moran's *I* | 0.8684 | 0.8440 | 0.9364 | 0.8263 | 0.8847 | 0.6293 | 0.8279 |
| Z-score | 307.13 | 187.77 | 234.60 | 243.20 | 306.89 | 80.79 | 168.25 |
| *p*-value | 0.0000 | 0.0000 | 0.0000 | 0.0000 | 0.0000 | 0.0000 | 0.0000 |

### 4.3.2. Local Spatial Autocorrelation Analysis

The local spatial autocorrelation analysis of SFQ grade can accurately determine the spatial location where the study objects aggregate or disperse, and the results can be represented by spatial LISA distribution diagram. In LISA cluster map, HH aggregation (high–high type) and LL aggregation (low–low type) indicate that the SFQ grades have higher spatial aggregation characteristic (positive correlation type), while HL variation (high–low type) and LH variation (low–high type) indicate a negative spatial correlation. Hence, the region has a discrete characteristic. Figure 5 shows the LISA cluster map of local spatial autocorrelation of SFQ grades in Yunnan. It can be seen from Figure 5 that the LISA cluster types of SFQ grades in Yunnan include HH aggregation (high–high type), LL aggregation (low–low type), the HL variation (high–low type), LH (low–high type) and non-significance. Among them, the HH aggregation (high-high type) and LL aggregation (low–low type) are dominant, indicating that the SFQ grades have higher spatial aggregation characteristics. The SFQ grade characterized by HH aggregation (high–high type) widely distributes, mainly in the Southwest Yunnan, Northeast Yunnan and Northwest Yunnan, with only scattered distribution in other regions. The SFQ grades of evaluation units corresponding to the above regions are relatively lower. Likewise, the SFQ grade characterized by LL aggregation (low-low type) also has a wide distribution area, mainly distributing in central Yunnan, Southern Fringe and Southeastern Yunnan, and with only scattered distribution in other regions. The SFQ grades in these regions are relatively higher. The SFQ grades characterized by HL outlier (high-low type) and LH outlier (low-high type) have smaller distribution area, with only scattered distribution in each region. The non-significance type of SFQ grade mainly distributes in the Southern Fringe and is distributed in a scattered manner in other regions.

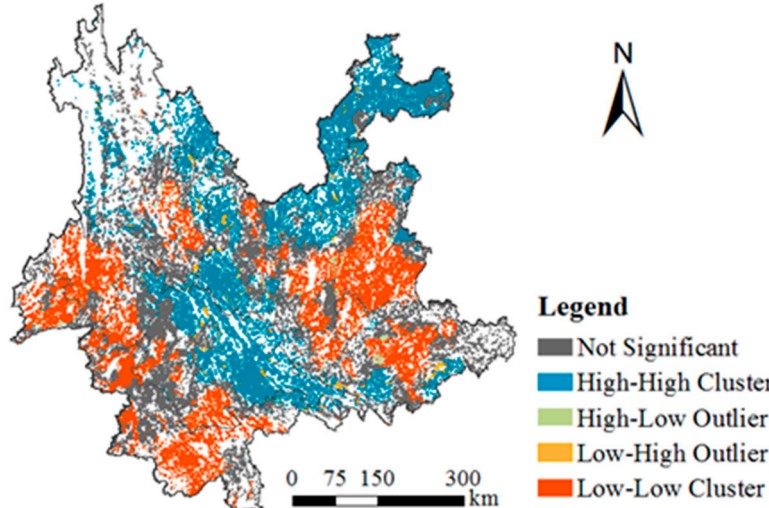

**Figure 5.** Local spatial autocorrelation local indicators of spatial association (LISA) cluster map of SFQ grade.

### 4.4. Characteristics of Space Cold and Hot Spots in SFQ

The global spatial autocorrelation analysis could reveal the overall correlative structure characteristics of SFQ grades of different evaluation units, but it can't be used to measure the spatial correlation mode between elements or attributes of adjacent regions [50]. Hence, the spatial cold and hot spots analysis tools are needed to further explore the correlation degree of SFQ grades in local regions. In this study, the spatial distribution of cold and hot spots are obtained by analyzing the cold and hot spots for the SFQ grades in the county-level region in Yunnan, as shown in Figure 6. Obviously, the hot spot regions of SFQ grades (high-value region of SFQ grade) are mainly distributed in the central Yunnan and Southern Fringe. In these regions, the slope farmland has a small slope, high

continuity degree, and light soil erosion degree. Moreover, the cultivation utilization level of slope farmland is higher, and the SFQ grade is higher. So these regions are the high-value regions of SFQ in Yunnan, as well as the key protection regions of SFQ.

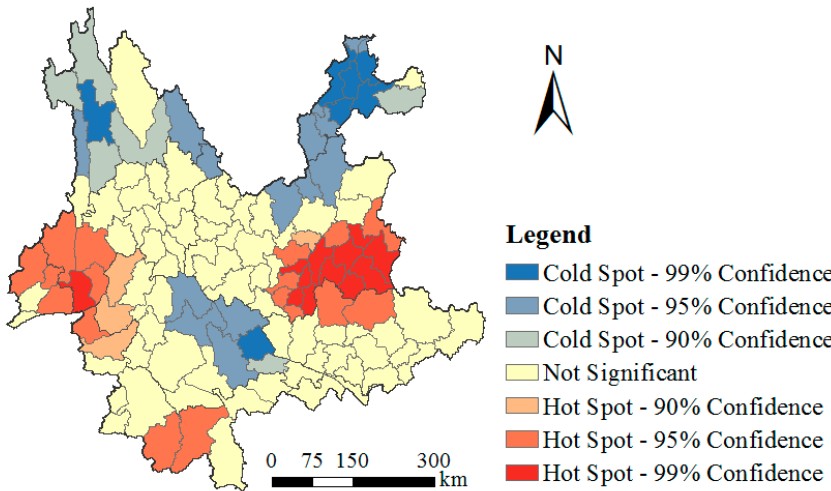

**Figure 6.** Cold hot spot analysis results of quality grades in Yunnan slope farmland.

The cultivation utilization level of slope farmland are maintained or improved by various land management measures. The cold spot region of SFQ mainly distributes in the Northeastern Yunnan, Northwestern Yunnan, and Southwestern Yunnan. In these regions, the slope farmland has high fragmentation degree, large slope classification and serious soil erosion degree of slope surface. Furthermore, the unreasonable cultivation utilization mode results in the low SFQ grade and the difficulty in the effective performance of land productivity. The SFQ in these areas is preferentially adjusted and controlled. The SFQ adjustment measures would be implemented to further improve the SFQ level. The other regions belong to the non-significance type, and their SFQ grade is medium. Simultaneously, they are the key regions for the SFQ adjustment. Thus, the SFQ level would be further improved by the adjustment measures, so as to make the SFQ grade distribution type extend to the hot spot region.

## 5. Conclusions

With the increasingly prominent problem of farmland quality degradation, farmland quality evaluation has become a hot issue, but there are few studies on SFQ evaluation and its spatial pattern of different scales. In this paper, taking Yunnan Province, a typical plateau mountain area, as an example, through the establishment of the SFQ evaluation system, the quality of slope farmland in Yunnan Province was quantitatively evaluated, and the spatial structure characteristics of SFQ were analyzed. The main conclusions are as follows.

(1) The SFQ indexes in Yunnan Province distributes between 0.36 and 0.81, with a mean of 0.59 ± 0.06. The SIFIs of most evaluation units are less than 0.6. The spatial distributions of SFQ indexes are significantly different. Moreover, the SFQ grade is based on sixth-class, fifth-class, seventh-class and fourth-class land. The SFQ grade is relatively higher in the Southern Fringe, central Yunnan, Western Yunnan and Southeastern Yunnan, while the SFQ grade is relatively lower in Northeastern Yunnan and Northwestern Yunnan.

(2) The SFQ indexes present a normal spatial distribution, and the Gaussian model can well fit the semi-variance function of SFQ indexes. The Nugget $C_0$ of spatial distribution of SFQ indexes is 0.5293, and the Nugget coefficient $C_0/(C_0+C)$ is 53.97%. Furthermore, the spatial distribution of SFQ indexes is moderately autocorrelated. The structural factors, such as climatic conditions,

soil property, moisture conditions, spatial morphology, etc., play a major role in the spatial heterogeneity of SFQ indexes, but the influence of random factors should not be ignored.

(3)  The Moran's *I* value of global spatial autocorrelation of SFQ grades is 0.8489. The spatial distribution of SFQ grades has a significant spatial aggregation characteristic. The spatial autocorrelation types of SFQ grades are based on HH aggregation and LL aggregation, as well as the types of their LISA cluster are based on HH aggregation and LL aggregation.

(4)  The cold spot and hot spot distributions of SFQ grades display the significantly spatial difference. The hot spot area is mainly distributed in the Central Yunnan and the Southern Fringe, while the cold spot area is mainly distributed in the Northeastern Yunnan, Northwestern Yunnan and Southwestern Yunnan.

The spatial distribution of SFQ grades in Yunnan Province has significant spatial aggregation characteristics, which provides a scientific basis for the management of slope farmland. The central Yunnan and Southern Fringe are high value areas of SFQ distribution in Yunnan, and their spatial autocorrelation type is mainly LL and is the core protection area of SFQ in Yunnan. It is necessary to implement various land management measures to maintain or improve the cultivation and utilization level of slope farmland. Northwest Yunnan, Northeast Yunnan and Southwest Yunnan are low value areas of SFQ distribution, and their spatial autocorrelation type is mainly H–H, which is the priority area of SFQ regulation. Therefore, it is necessary to further improve the quality level of slope farmland by implementing SFQ regulation measures. Western Yunnan and Southeast Yunnan are the middle value areas of SFQ distribution, and their spatial autocorrelation and cold hot spot distribution types are not significant, which are the key areas of SFQ regulation. It is necessary to further improve the quality level of slope farmland and extend the distribution type of SFQ grade to hot spot area through regulation measures.

**Author Contributions:** Conceptualization, Z.C. and D.S.; Investigation, Z.C.; Methodology, Z.C.; Resources, D.S.; Supervision, D.S.; Writing—Original draft, Z.C. All authors have read and agreed to the published version of the manuscript.

**Funding:** This work has been financially supported by the Public welfare industry research project (agriculture): establishment of evaluation index system for rational tillage in slope farmland (No:201503119-01-01).

**Acknowledgments:** Some of the data have been provided by the resource and environment data cloud platform (http://www.gscloud.cn). All assistance received has been greatly appreciated.

**Conflicts of Interest:** The authors declare no conflict of interest.

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
