# Peer review of "Spatial Structure Characteristics of Slope Farmland Quality in Plateau Mountain Area: A Case Study of Yunnan Province, China"

_sustainability, doi:10.3390/su12177230_

Round 1

Reviewer 1 Report

In general, the article is reasonably well written. Its layout is legible and appropriate. However, I have a few comments:

1. What are the boundaries of Western Yunnan? In figure 1 we have a marked area that has no name. Is there a mistake in naming this area and the lack of highlighting it in the calculations, or is there an error in incorrectly drawing borders for Western Yunnan. And if the latter, what about the calculation results - it related to the whole of this region or only to the marked part?

2. In Results and Discussion, please correct the names of the sub-items. Their names should start with a capital letter - but that's a technical note. More importantly, the discussion is virtually omitted in this section. In addition to references 50, 54 and 55) of the literature, barracks also refer to the results of other studies. This gap should be bridged. Do other studies conducted for this area confirm such spatial distribution of farmland quality? Does anyone examine these issues with methods other than those proposed in the article? What conclusions from these studies have been confirmed in this article? What new was discovered thanks to the autocorrelation analysis?

3. What I miss the most in this article are attempts to translate the conclusions of the study into sustainable development issues in the described area. What does this autocorrelation spatial arrangement (Figure 6) mean for development policy? What does he mean for the authorities? What for agricultural producers? In the introduction we have a reference to the three-in-one protection strategy in Yunnan Province, how do the article results translate into this strategy? In my opinion, the article would have gained significance if the conclusions also referred to these issues in the conclusions.

Author Response

Response to Reviewer 1 Comments

Point 1: What are the boundaries of Western Yunnan? In figure 1 we have a marked area that has no name. Is there a mistake in naming this area and the lack of highlighting it in the calculations, or is there an error in incorrectly drawing borders for Western Yunnan. And if the latter, what about the calculation results - it related to the whole of this region or only to the marked part?

Response 1:In Fig. 1, the area on the left of Western Yunnan does not show the name of the area in the map, but in fact this area and the area below the Southwest Yunnan constitute the Southern fringe area. The two areas mentioned above are not contiguous in the map, so they seem to be an independent division. This feature is considered in the calculation of partition data. In the process of modification, the partition of the area is marked. The modified partition is shown in Figure 1

Point 2:  In Results and Discussion, please correct the names of the sub-items. Their names should start with a capital letter - but that's a technical note. More importantly, the discussion is virtually omitted in this section. In addition to references 50, 54 and 55) of the literature, barracks also refer to the results of other studies. This gap should be bridged. Do other studies conducted for this area confirm such spatial distribution of farmland quality? Does anyone examine these issues with methods other than those proposed in the article? What conclusions from these studies have been confirmed in this article? What new was discovered thanks to the autocorrelation analysis?

Response 2:The sub headings of the results and discussions have been modified to ensure that the first letter of the title is capitalized. In the process of revision, the author analyzed the literature again, referred to the latest literature of farmland quality evaluation in recent years, and added the comparative discussion on the spatial distribution characteristics of farmland quality in different regions, so as to evaluate the conclusion confirmed by this study. This study is based on the quality of slope farmland. It systematically analyzes the quality evaluation and spatial distribution characteristics of slope farmland at provincial regional scale. The study confirms the spatial clustering distribution characteristics of slope farmland quality, and the research results can provide scientific basis for slope farmland quality management.

Point 3: What I miss the most in this article are attempts to translate the conclusions of the study into sustainable development issues in the described area. What does this autocorrelation spatial arrangement (Figure 6) mean for development policy? What does he mean for the authorities? What for agricultural producers? In the introduction we have a reference to the three-in-one protection strategy in Yunnan Province, how do the article results translate into this strategy? In my opinion, the article would have gained significance if the conclusions also referred to these issues in the conclusions.

Response 3:Spatial autocorrelation analysis is often used in the study of farmland quality zoning protection. At present, the spatial autocorrelation analysis of farmland quality has a wide application prospect in basic farmland division and high standard farmland construction. In the revision, we added the content of policy suggestions in the conclusion part (the revised part has been marked with red font) to provide reference for the administrative department to formulate the quality management policy of slope farmland. According to the spatial clustering distribution characteristics of slope farmland quality, we suggest to implement zoning management measures for slope farmland utilization in Yunnan, so as to improve the effectiveness of slope farmland resource management.

Reviewer 2 Report

I consider that the article is very interesting and has possibilities to be published in the journal.

However, there are a number of minor considerations that should be clarified

The abstract should be containing the methodology used in the analysis and, why is used it? Define LISA too.

In the paragraph 1 of the introduction the authors comment different problem of the farming in Chine (line 42-43) What of them induce to Chinese government to protection farming?

Although in the lines 124-125 say what this paper want to do, I recommend that the aim of the paper was clarified and comment, why is relevant?

Line 441, the I de Moran, how is defined the relationship distance? Inverse distance, inverse distance square, etc…and how is calculated the distance? Euclidean?

Results and discussion I think that the author should be introduced more studies to compared to the results obtained in the paper.

The conclusion is a resume of the results. The author should start by describing what has been done on the paper. Then, comment the more relevant obtain and the policy recommendation from the results obtain.

Author Response

Response to Reviewer 2 Comments

Point 1: The abstract should be containing the methodology used in the analysis and, why is used it? Define LISA too.

Response 1: Local spatial autocorrelation analysis is a common method in spatial autocorrelation analysis. In the revision of the abstract, we added the analytical methods used in the study and clarified the meaning of LISA (local indicators of spatial association).

Point 2: In the paragraph 1 of the introduction the authors comment different problem of the farming in Chine (line 42-43) What of them induce to Chinese government to protection farming?

Response 2: As a developing country with a population of 1.4 billion, food security is a major problem for China's sustainable development. However, at present, there are some problems in the utilization of farmland in China, such as the decline in the quality, the occupation of farmland, the fragmentation of spatial distribution, and the frequent occurrence of ecological problems. Therefore, the Chinese government has implemented a series of policies to protect cultivated land. Based on this background, this study evaluated the quality of slope farmland, analyzed the spatial distribution characteristics of slope farmland quality, and provided scientific basis for sustainable utilization of slope farmland. In order to make the background of topic selection more clear, the research background is revised in this revision.

Point 3: Although in the lines 124-125 say what this paper want to do, I recommend that the aim of the paper was clarified and comment, why is relevant?

Response 3: The purpose of this paper has been revised and perfected in the introduction.

Point 4: Line 441, the I de Moran, how is defined the relationship distance? Inverse distance, inverse distance square, etc…and how is calculated the distance? Euclidean?

Response 4: In spatial autocorrelation analysis, the spatial weight should be determined first. There are two methods to determine the spatial weight matrix, which are based on adjacency relationship and distance relationship. Referring to the existing research results, this paper uses the K-nearest method to determine the spatial weight。The above contents have been supplemented and improved in the paper.

Point 5: Results and discussion I think that the author should be introduced more studies to compared to the results obtained in the paper.

Response 5: The subtitles of the results and discussion sections have been modified so that the first letters of the titles are capitalized. In the process of revision, the author analyzed the literature again, referred to the latest literature of farmland quality evaluation in recent years, and added the comparative discussion on the spatial distribution characteristics of farmland quality in different regions, so as to evaluate the conclusion confirmed by this study. The revision has been marked in red in the paper.

Point 6: The conclusion is a resume of the results. The author should start by describing what has been done on the paper. Then, comment the more relevant obtain and the policy recommendation from the results obtain.

Response 6: The conclusion has been revised according to the review comments. In the conclusion part, the paper first describes the work done, evaluates the results of the study, and finally adds the content of policy recommendations to provide reference for administrative departments to formulate slope farmland quality management policies. According to the spatial clustering distribution characteristics of slope farmland quality, we suggest to implement zoning management measures for slope farmland utilization in Yunnan, so as to improve the effectiveness of slope farmland resource management.
